# Status of RadioMonteCarLow and Strong2020 $e^+e^-$ database activities

Graziano Venanzoni[1,2]⋆

**1** University of Liverpool, Liverpool L69 3BX, U.K.
**2** INFN, Sezione di Pisa, Pisa, Italy
* graziano.venanzoni@liverpool.ac.uk

December 14, 2024

## Abstract

**During the last 15 years the Radio MonteCarLow Working Group has been providing valuable support to the development of radiative corrections and Monte Carlo generators for low energy $e^+e^-$ data and $\tau$-lepton decays. While the working group has been operating for more than 15 years without a formal basis for funding, parts of our program have recently been included as a Joint Research Initiative in the group application of the European hadron physics community, STRONG2020, to the European Union, with a more specific goal of creating an annotated database for low-energy hadronic cross sections in $e^+e^-$ collisions. We will report on both these initiatives.**

# 1 Introduction

The importance of continuous and close collaboration between the experimental and theoretical groups is crucial in the quest for precision in hadronic physics. This is the reason why the Working Group (WG) on "Radiative Corrections and Monte Carlo Generators for Low Energies" (Radio MonteCarLow, see http://www.lnf.infn.it/wg/sighad/) was formed a few years ago bringing together experts (theorists and experimentalists) working in the field of low-energy $e^+e^-$ physics and partly also the $\tau$-lepton community. Its main motivation was to understand the status and the precision of the Monte Carlo generators (MC) used to analyze the hadronic cross section measurements obtained in energy scan experiments as well as with radiative return method, to determine luminosities. Whenever possible specially prepared comparisons, *i.e.* comparisons of MC generators with a common set of input parameters and experimental cuts, were performed within the project. The main conclusions of this major effort were summarized in a report published in 2010 [1]. During the years the WG structure has been enriched of more research topics including seven subgroups: Luminosity, R-measurement, ISR, Hadronic VP g-2 and Delta alpha, gamma-gamma physics, FSR models, tau decays. The working group had been operating for more than 15 years without a formal basis and a dedicated funding. Recently parts of the program have been included as a Joint Research Initiative (JRA3-PrecisionSM) in the group application of the European hadron physics community, STRONG2020 (http://www.strong-2020.eu), to the European Union, with a more specific goal of creating an annotated database for the low-energy hadronic cross section data in $e^+e^-$ collisions. The database will contain information about the reliability of the data sets, their systematic errors and the treatment of Radiative Corrections. All these efforts have been recently revitalized by the new high-precision measurement of the anomalous magnetic moment of the muon [2], which shows a $4.2\sigma$ discrepancy with respect to the state-of-the-art theoretical prediction from the Standard Model (SM) [3]. Moreover the recent high precision lattice evaluation of the BMW collaboration [4] shows tension with the time-like data-driven determinations of $a_\mu^{HLO}$, being $2.2\sigma$ higher than the Muon $g-2$ Theory Initiative data-driven value. In view of the experimental efforts under way at Fermilab (USA) and J-PARC (Japan) to improve the $a_\mu$ accuracy, a consolidation (and a possible improvement) of its SM prediction is extremely important. Here we discuss the status of the radiative corrections and Monte Carlo Tools for low energy $e^+e^-$ data and the prospects towards the full NNLO MC calculation.

# 2 Hadronic Vacuum Polarization to the Muon g-2 (and $\alpha_{em}(M_{Z^0})$)

The Theory prediction for the muon magnetic anomaly is limited by vacuum fluctuations involving strongly interacting particles. They mainly originate from the leading hadronic vacuum polarization term $a_\mu^{HLO}$ which cannot be reliably calculated perturbatively in QCD, due to the non-perturbative nature of the strong interactions at low energy. It is possible to overcome this problem by means of a dispersion relation technique involving experimental data measuring the cross section of electron positron annihilation into hadrons, $e^+e^- \rightarrow hadrons$, (so-called "time-like" or "dispersive" approach):

$$
\begin{aligned}
a_\mu^{\text{HLO}} &= \frac{1}{4\pi^3} \int_{m_\pi^2}^{\infty} ds\, K(s)\sigma^0(s) \\
&= \frac{\alpha^2}{3\pi^2} \int_{m_\pi^2}^{\infty} ds\, K(s) R_{had}(s)/s \,.
\end{aligned}
\tag{1}
$$

where $R_{had}(s)$ is the ratio of the total $e^+e^- \to hadrons$ and the Born $e^+e^- \to \mu^+\mu^-$ cross sections in the pointlike ($m_\mu = 0$) limit, $K(s)$ is a smooth function and $m_\mu(m_\pi)$ is the muon (pion) mass. The functional form of the integral emphasizes low energy contributions where the cross section $e^+e^- \to hadrons$ is densely populated with resonances and modulated by threshold effects. This makes the dispersive approach evaluation of $a_\mu^{HLO}$ highly challenging with an error dominated by systematic uncertainties of data [3].

For the quest of precision in the calculation of $a_\mu^{HLO}$ it's important to understand the correctness of Eq. (1) and how it is related to $e^+e^-$ annihilation cross sections into hadrons data. Equation (1) was obtained from the fundamental assumptions of causality and unitarity; therefore, it can be considered exact in the relevant order of perturbation theory. All diagrams in which the virtual photon couples to one or more hadrons and an arbitrary number of other particles must be included. Processes with several photon exchanges or photon emission by initial particles should not be taken into account in $R_{had}(s)$; however, they cannot be completely excluded from the observed processes using selection criteria. Therefore, the observed hadron production cross sections should be supplemented with radiative corrections. Processes with photon emission by final hadrons should be taken into account in $R_{had}(s)$. $R_{had}(s)$ corresponds to one-particle-irreducible insertions in the photon propagator, and therefore hadron production should proceed via one virtual photon without loop insertions. In the observed process $e^+e^- \to hadrons$ the virtual photon is described by the full propagator with the vacuum polarization taken into account. Therefore, the vacuum polarization contribution should be excluded from the observed cross section. The measured hadron cross sections are typically normalized to the Bhabha cross section or $\mu\mu$ cross section. The first case requires the knowledge of the machine luminosity, while in the latter case the vacuum polarization contribution is automatically canceled.

Usually the computation of $a_\mu^{HLO}$ is done by using experimental data in the low-energy range and perturbative QCD at higher energy. The energy $E_{cut}$ varies in different calculations. In the region of relatively high energies, inclusive measurements of the cross section are carried out while at lower energies $< 2$ GeV, the exclusive measurement of cross sections of each separate final hadron state $e^+e^- \to 2\pi; 3\pi; 4\pi; 2K; ...$ is mainly used, and $a_\mu^{HLO}$ is calculated as a sum of contributions due to individual final hadron states. For a long time, VEPP-2M at Novosibirsk was the only $e^+e^-$ collider capable of working at low energies. On this collider, which operated from 1975 until 2000, a series of measurements of hadron cross sections was performed at low energies mostly by CMD-2 and SND experiments which pioneered the measurements of hadronic cross sections by changing the collider energy (energy scan). The last 20 years have seen a big effort on $e^+e^-$ data in the low energy region. More refined data were produced for experiments at energy-scan colliders (CMD3 and SND at VEPP-2000 and BESIII at BEPC colliders) and at flavor factories, where the use of Initial State Radiation (ISR), pioneered by the KLOE and BaBar experiments, opened a new way to precisely obtain the $e^+e^-$ annihilation cross sections into hadrons at particle factories operating at fixed beam energy. New dedicated tools and refined theoretical treatment were developed for the analysis of data [1]. All this effort led to a substantial reduction on the uncertainty of $a_\mu^{HLO}$ to 0.6%.

## 3   Radiative corrections

The precise determination of the hadronic cross sections (accuracy $\lesssim 1\%$) requires an excellent control of higher order effects like Radiative Corrections (RC) and the non-perturbative hadronic contribution to the running of $\alpha$ (i.e. the vacuum polarisation, VP) in Monte Carlo (MC) programs used for the analysis of the data. Particularly in the last years, the increasing precision reached on the experimental side at the $e^+e^-$ colliders (VEPP-2M, VEPP-2000,

| MC generators for exclusive channels (exact NLO + Higher Order terms in some approx) | | | |
|---|---|---|---|
| MC generator | Channel | Precision | Comment |
| MCGPJ (VEPP-2M, VEPP-2000) | $e^+e^- \to e^+e^-; \mu^+\mu^-; \pi^+\pi^-; ...$ | 0.2% | photon jets along all particles (collinear Structure function) with exact NLO matrix elements |
| BabaYaga@NLO (KLOE, BaBar, BESIII) | $e^+e^- \to e^+e^-; \mu^+\mu^-; \gamma\gamma$ | 0.1% | QED Parton Shower approach with exact NLO matrix elements |
| BHWIDE (LEP) | $e^+e^- \to e^+e^-$ | 0.1% | Yennie-Frautschi-Suura (YFS) exponentiation method with exact NLO matrix elements |

Table 1: MC generators for exclusive channels.

DAΦNE, BEPC, PEP-II and KEKB) led to the development of dedicated high precision theoretical tools: BabaYaga (and its successor BabaYaga@NLO) for the measurement of the luminosity, MCGPJ for the simulation of the exclusive QED channels, and PHOKHARA for the simulation of the process with Initial State Radiation (ISR) $e^+e^- \to hadrons + \gamma$, are examples of MC generators which include NLO corrections with per mill accuracy. In parallel to these efforts, well-tested codes such as BHWIDE (developed for LEP/SLC colliders) were adopted.

Theoretical accuracies of these generators were estimated, whenever possible, by evaluating missing higher order contributions. From this point of view, the great progress in the calculation of two-loop corrections to the Bhabha scattering cross section was essential to establish the high theoretical accuracy of the existing generators for the luminosity measurement. However, usually only analytical or semi-analytical estimates of missing terms exist which don't take into account realistic experimental cuts. In addition, MC event generators include different parametrisations for the VP which affect the prediction (and the precision) of the cross sections and also the RC are usually implemented differently.

These arguments evidently imply the importance of comparisons of MC generators with a common set of input parameters and experimental cuts. Such *tuned* comparisons, which started in the LEP era, are a key step for the validation of the generators, since they allow to check that the details entering the complex structure of the generators are under control and free of possible bugs. This was the main motivation for the *"Working Group on Radiative Corrections and Monte Carlo Generators for Low Energies" (Radio MontecarLow)*, which was formed a few years ago bringing together experts (theorists and experimentalists) working in the field of low energy $e^+e^-$ physics and partly also the $\tau$ community.

In addition to tuned comparisons, technical details of the MC generators, recent progress (like new calculations) and remaining open issues were also discussed in regular meetings.

## 4   Prospects on RCs and MC tools

With a lot of new data from VEPP-2000, BaBar, BelleII, and BESIII experiments with better quality and refined systematic errors, Radiative correction and MC generators (including ISR)

| MC generators for ISR (from approximate to exact NLO) | | | |
|---|---|---|---|
| MC generator | Channel | Precision | Comment |
| EVA (KLOE) | $e^+e^- \to \pi^+\pi^-\gamma$ | O(%) | Tagged photon; ISR at LO + Structure Function; FSR: pointlike pions |
| AFKQED (BaBar) | $e^+e^- \to \pi^+\pi^-\gamma$ | Depends on event selection (can be as good as PHOKHARA) | ISR at LO + Structure Function; FSR: pointlike pions |
| PHOKHARA (KLOE, BaBar, BESIII) | $e^+e^- \to \pi^+\pi^-\gamma; \mu^+\mu^-\gamma; 4\pi\gamma; ....$ | 0.5% | ISR and FSR (sQED+Form Factor) at NLO |
| KKMC | $e^+e^- \to l^+l^-\gamma$ | High accuracy (< 1%) only for lepton pairs | Yennie-Frautschi-Suura (YFS) exponentiation for soft photons + hard part and subleading terms in some approximation |

Table 2: MC generators for ISR.

should aim at 0.1% uncertainty. Tables 1 and 2 show the status of MC generators for exclusive channels and for ISR respectively. All these generators have matrix elements which go from approximate to exact NLO. To reach 0.1% uncertainty a NNLO generator would be required. Although very challenging, this goal in accuracy may benefit from recent developments [5]. It would be also important to have under control at the required accuracy the FSR model [5].

# 5  Strong2020

Recently parts of the program have been included as a Joint Research Initiative (JRA3-PrecisionSM) in the group application of the European hadron physics community, STRONG2020 (http://www.strong-2020.eu), to the European Union, with a more specific goal of creating an annotated database for the low-energy hadronic cross section data in $e^+e^-$ collisions. The database will contain information about the reliability of the data sets, their systematic errors and the treatment of Radiative Corrections.

The PrecisionSM database is one of the specific objectives of Strong2020, a European joint research initiative that groups together researchers from different scientific frontiers (low energy, high energy, instrumentation and infrastructures) with the broad goal of study strong interactions and develop applications for beyond fundamental physics.

The plan for implementing the precisionSM database is summarized in the following steps. The first step consists in collecting, with the help of experts, the list of published precision low energy data from various experiments. Each of these measurements is then catalogued and made available in a public repository called HEPData (https://www.hepdata.net/). A coordinator, usually a point-of-contact person designated by each experiment or project, uploads the data in the repository and appoints a reviewer to perform a cross-reference validation. Validated data are then made public and indexed in an easy, transparent and accessible way

through the PrecisionSM database webpage [6].

At present, we are collecting measurements from the $e^+e^- \to \pi^+\pi^-$ channel, which is an important channel for computing the muon $g-2$ hadronic vacuum polarization term, from the following experiments: BaBar (SLAC, USA); Beijing Spectrometer (BESIII, China); KLOE (LNF, Italy); SND, TOF, OLYA, VEPP, CDM, CMD-2 and CMD-3 (Novosibirsk, Russia); MEA and BCF (Adone LNF, Italy), CLEO (Cornell Electron Storage Ring, USA). These datasets are in the process to be uploaded and validated in HEPData. Measurements details are also being collected in the PrecisionSM website. The plan for the future is to catalogue measurements for other $e^+e^- \to$ hadrons channels as well as develop tools to perform data downloading and simple elaboration, e.g., comparative plots.

## 5.1 An annotated database (PrecisionSM) of low energy $e^+e^-$ hadronic cross section measurements

The PrecisionSM annotated database design relies on the HEPData (https://www.hepdata.net/) web database to store the experimental data for some precision tests of the Standard Model and on a custom web site to list the measurements with links to their HEPData location together with information on how to perform elaborations with them. HEPData is an open-access repository for scattering data from experimental particle physics, which is regularly used to store published experimental results and additional information on them, typically tabular data published as supplemental information. HEPData is used extensively by the LHC experimental collaborations and with diverse frequency by several other collaborations. The web page that hosts the experimental data links their publication on InspireHEP, and InspireHEP provides links to HEPData. Metadata are used to classify the measurements and to define and list subsets of them. A static web site (https://precision-sm.github.io/) is used to store for each measurement of interest a link to its web page on HEPData and additional information on how to use the stored data. The web site is built with the Nikola static web site generator by compiling web page sources in user-friendly markup languages like Markdown. All sources are hosted in a Git repository on Github.com, which provides versioning and a reliable collaborative framework. The Nikola generator organizes the web pages in a hierarchical tree of categories. Each page has one category tree location plus multiple tags, which provide the ability to group the measurements as desired. Measurements of the same quantity are collected in responsive plots using the jsRoot Javascript package. These plots provide the ability to interactively manipulate the plots, to show the experimental data for each data point and to select which publications are included, following the example of the muon g-2 hadronic vacuum polarization web site built by Fedor Ignatov (https://cmd.inp.nsk.su/~ignatov/vpl/). In order to get all experimental measurements of interest for PrecisionSM in HEPData, an authorized coordinator has been chosen with the required privileges to submit measurements to HEPData. The completeness and accuracy of the measurements in HEPData will be checked and reported in the Nikola web site. All relevant experimental collaborations are being solicited to submit to HEPData all the measurements of interest for PrecisionSM that are presently missing.

## 6 Conclusion

The importance of continuous and close collaboration between the experimental and theoretical groups is crucial in the quest for precision in hadronic physics. That was done in the last 15 years within the Working Group on Radiative Corrections and Monte Carlo Generators for Low Energies (Radio MontecarLow). That effort resulted in an accuracy on MC generators for low energy $e^+e^-$ processes between 0.2 and 0.5%. New data and improved evaluation of $a_\mu^{HLO}$ re-

quires improvement on MC generators at 0.1% which would require a NNLO matrix elements calculation. That accuracy although challenging can benefit from recent developments [5]. In addition to the request in accuracy, for the better understanding of the experimental data and to facilitate the information, the Strong2020 project will contribute with an annotated database for low energy hadronic cross sections in $e^+e^-$ collisions, which will contain information about the reliability of the data sets, their systematic errors, and the treatment of RC.

## Acknowledgements

This work was supported by the European Union STRONG 2020 project under Grant Agreement Number 824093 and by the Leverhulme Grant LIP-2021-014.

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
