# Peer review of "Status of Radio MonteCarLow and Strong2020 $e^+e^-$ database activities"

_SciPost Physics Proceedings_

## Round 1 · Referee Report · Denis Epifanov (Referee 1) · 2022-4-5

Report

The paper describes the activity of the working group on the
"Radiative Corrections and Monte Carlo Generators for Low Energies" (Radio MontecarLow) during last 15 years. The Radio MontecarLow group efforts resulted in the development and cross checks of the high accuracy MC generators for low energy e+ e- processes. While the accuracy of the MC generators (BHWIDE, BabaYaga) for QED processes, like e+ e- -> e+ e-, mu+ mu-, gamma gamma, already reaches 0.1% level, there is a big challenge to improve the existing MC generators (EVA, AFKQED, PHOKHARA, MCGPJ) for the e+ e- -> hadrons processes, like e+ e- -> pi+ pi- (gamma), and reach similar 0.1% level. For that heavy NNLO calculations are needed for the processes under interest taking properly the effects of ISR, FSR and multiloop virtual corrections.
The other activity of the Radio MontecarLow group have been included as a Joint Research Initiative in the application of the European hadron physics community to the European Union (STRONG2020). It concerns the creation and support of an annotated database for low-energy hadronic cross sections in e+ e- collisions.
The paper discusses the motivation to create the PrecisionSM database
(evaluation of the hadronic contribution to the anomalous magnetic moment of muon), as well as briefly describes the structure and main tools of PrecisionSM.

Requested changes

1) page 3, line 6 up from the end of Sec.2:

  VEP-2000 --> VEPP-2000

2) page 3, line 7 up from the end of Sec.2:

  The sentence:

  "Better data were produced at fixed energy ... at fixed beam energy."

   can be misleading for the uninformed reader.

   Namely, CMD3 and SND at VEPP-2000 are ENERGY SCAN experiments, 
   they don't stand and collect statistics at the fixed beam energy like KLOE 
   or  BABAR did. 
   Instead, CMD-3 and SND collect the data in the wide c.m.s. 
   energy range 2E = 0.3 -- 2.0 GeV.

   I'd suggest author to write this sentence more clear.

3) page 3, line 2 from the beginning of Sec.3:

  Please, add VEPP-2000 to the mentioned list of e+ e- colliders: VEPP-2M, 
  DAFNE, BEPC, PEP-II, KEKB.

4) page 5, Table 2:

  EVA string:  e+ e- pi+ pi- gamma --> e+ e- -> pi+ pi- gamma (arrow is 
  missed) 
  also here:  O(%) -> O(1%)

  AFKQED string:   "Depends on event section" - unclear 
                                              may be "Depends on event selection" ?

  Is it possible to specify at least the diapason of precisions 
  for this generator ?

  KKMC string: The claim: "High accuracy only for muon pairs" is unclear ?

 What do you mean here ? The e+ e- -> lep+ lep- gamma production cross 
 section only or you also include the accuracy of the simulation of the 
 further lep+/lep- decays ?  If only lep+ lep- production is implied, 
 so the e+ e- -> tau+ tau- gamma has the same accuracy as 
 e+ e- -> mu+ mu- gamma and it is better than 1%.

 I think it is reasonable for KKMC to put "<1%" for the accuracy of the 
 simulation of e+ e- -> lep+ lep- gamma, lep = mu, tau.

  • validity: high
  • significance: high
  • originality: high
  • clarity: high
  • formatting: excellent
  • grammar: perfect

Author:  Graziano Venanzoni  on 2022-07-03  [id 2626]

(in reply to Report 1 by Denis Epifanov on 2022-04-05)

Thanks very much for the careful reading, I agree with most of the comments of the referee.
about: "EVA string: e+ e- pi+ pi- gamma --> e+ e- -> pi+ pi- gamma (arrow is missed)
also here: O(%) -> O(1%) "
it's unclear if precision of EVA is O(1%). I would think more of few % but of course it depends also on the selection cuts

about the question "KKMC string: The claim: "High accuracy only for muon pairs" is unclear ? " I will ask the authors of KKMC.

I will wait for an OK from the editor and produce a new report with the comments from the referee.

---

## Editorial Decision

accepted_in_target_journal